# Time–Motion Analysis of the 2023 Women’s World Boxing Championships Finals

**DOI:** 10.3390/sports13060187

**Published:** 2025-06-17

**Authors:** Francesca Martusciello, Andrea Perazzetti, Arben Kaçurri, Marco Consolati, Antonio Tessitore

**Affiliations:** 1Department of Movement, Human and Health Sciences, University of Rome “Foro Italico”, 00135 Rome, Italy; perazzettiandrea@gmail.com (A.P.); antonio.tessitore@uniroma4.it (A.T.); 2National PhD Programme in Kinesiology and Sport Sciences, Department of Neurosciences, Biomedicine and Movement, University of Verona, 37129 Verona, Italy; 3Sports Research Institute, Sports University of Tirana, 1001 Tirana, Albania; akacurri@ust.edu.al; 4National Teams Technical Commission, Italian Boxing Federation (FPI), 00196 Rome, Italy; marco.consolati@fpi.it

**Keywords:** match analysis, time–motion analysis, amateur boxing, female boxing, combat sport, fighting

## Abstract

Background: This study investigated boxers’ activity profiles during the final matches (3 × 3 min format) of the IBA 2023 Women’s World Boxing Championships. Methods: Footage of the 12 finals was used to analyse the frequency and duration of fighting (F), punching (P), clinching (C), no-fighting (NF), and arbitral interruption (AI) phases. The analysis was conducted both for weight categories and divisions (lightweight (LWC): 48, 50, 52, 54, 57, and 60 kg; middleweight (MWC): 63, 66, 70, and 75 kg; heavyweight (HWC): 81 and 81+ kg). Results: Pooled data per round revealed significant differences for P (<0.001), C (*p* = 0.002), NF (*p* < 0.001), and AI (*p* < 0.001) phases, as well for P mean duration across rounds (*p* < 0.001). The MWC division showed significantly shorter F duration compared with the LWC (*p* = 0.007) and MWC divisions (<0.001). The F/NF total time ratio showed a prevalence of F in the 48, 50, 63, and 81+ kg categories, while NF prevailed in the 54, 57, 60, and 75 kg categories. Conclusions: While HWC primarily relied on C actions, the 54, 57, 60, and 75 kg categories showed higher NF frequency. This fact explains a different strategic match management approach with deliberate rhythm, controlled pauses, and opponent analysis, which coaches should consider for enhancing athletes’ performance by considering weight categories.

## 1. Introduction

Boxing can be considered a full-contact combat sport characterised by the primary goal of effectively scoring punches on the opponent while minimising the risk of receiving punches in return [1], which requires boxers to have well-developed technical and tactical skills as well as a high level of physical and physiological fitness [2]. Even though amateur boxing, practiced by both males and females, may look like professional boxing, it has different rules and scoring systems, which means that the technical requirements for performance might change depending on the type of event [3] and that amateur boxers are less likely to get certain injuries [4]. From a technical perspective, the activity profile of a boxing match can be divided into fighting (F), no-fighting (NF), and arbitral interruption (AI) phases. In turn, the fighting phase, characterised by high-intensity activity, is divided into two sub-phases: punching (P) and clinching (C). The punching phase involves rapid, high-intensity strikes that require short-distance locomotion while performing offensive and defensive actions, while clinching (C) occurs when the fighters close the distance and lock with each other, reducing the space and attempting to control the opponent’s movements.

These features require the intervention of both aerobic and anaerobic energy systems [5], determining the intermittent nature of boxing [6]. In terms of the work/rest ratio, the high variability of performance in a boxing match is also influenced by the match’s formal organisation into rounds (with breaks in between) and the informal presence of micropauses and pauses for arbitral interruptions, the amount and duration of which are determined by the internal logic of the boxers’ interaction and not established by regulations [7]. For this reason, considering that the individual performance profiles are affected by the type and number of micropauses and phases occurring within the rounds, more research is needed to better understand this timing in relation to the different competitive levels and weight classes in male and female amateur boxing [8]. Such an approach would also help coaches to better plan their training.

As a non-invasive performance analysis technique, the use of time–motion analysis (TMA), by quantifying the mode, frequency, and duration of discrete activities, better reveals the effort/pause ratio structure of a round/match in striking combat sports, as in other sports. In boxing, a study involving ten male amateur boxers conducted by Davis et al. [5] found that most of their metabolic profiles were predominantly aerobic. These findings emphasise how aerobic rephosphorylation, which is needed to sustain a high activity rate during rounds, and the recovery of the high-energy phosphate system during breaks are two key factors in succeeding in boxing. Regarding the 3 × 3-min format, two studies examined the boxing competitions during the 2012 Olympic Games (London, UK). A study conducted by Davis et al. [9] was the first to provide a detailed activity profile of male Olympic-level boxing in the 3 × 3-min format. The authors observed that the total stop time increased with the progression of the rounds, the activity-to-break ratio decreased between the first and third rounds, and the activity rate increased progressively from the first to the other two rounds. Furthermore, minimal variations were discerned across the light-, middle-, and heavyweight divisions. Then, a study investigating the activity profiles of winning male boxers during the 2012 Olympic Games [10], which measured the punching frequency rate across the three rounds as well as the footwork duration, showed that winners exhibited a different activity profile, with a higher punching frequency and more footwork compared with losers. However, the time–motion analysis provided in boxing research has mainly examined the performance profile in men’s amateur boxing with different formats of number and duration of rounds, while scarce investigation has been conducted in women’s competitions.

A pivotal point in the evolution of women’s boxing transpired in 2009, when the International Olympic Committee (IOC) declared the inclusion of women’s boxing in the Olympic Games for the first time in 2012 [11]. To the best of our knowledge, only Davis et al. [12] investigated the activity profiles of elite women’s amateur boxing during the same 2012 Olympic Games to make a comparison with their male counterparts. The analysis of the footage in this study showed that rounds averaged 132 s because of referee interventions, with an overall activity-to-interruption ratio of 6.6:1, excluding breaks between rounds. The authors also noted increases in the frequency of interruptions and clinching durations in the final third round compared with the initial round. Clinch durations constituted most of the interruption time, while referee-initiated interruptions remained consistent at around 12 s throughout subsequent rounds [12].

Furthermore, women’s boxing consists of 12 weight categories, ranging from “Minimum” (48 kg) to “Heavy” (81+ kg). However, until the Paris 2024 Olympics, only six of these categories were included in the Olympic program.

Nevertheless, it is important to note that this study was conducted based on the previous 4 × 2-min format used until 2019 in female boxing for all kinds of international competitions (i.e., European, World, and Olympic), while it was later set in a 3 × 3-min format with a 1-min rest interval [13]. These two formats differ in both total duration and effort/recovery balance. A review by Chaabène et al. [2] compared the male 3 × 3-min format and the female 4 × 2-min format, showing the former involved a higher work-to-rest ratio (9:2) compared to the latter (8:3), possibly imposing a greater physiological strain.

To date, no study has comprehensively analysed the time–motion structure of elite women’s boxing using the 3 × 3-min format across all official weight categories. Based on the existing literature in male boxing and preliminary findings in female competitions, we hypothesised that the frequency and duration of fighting, no-fighting, and arbitral interruption phases would significantly vary both across weight categories and rounds. Therefore, this study aimed to provide an activity profile of the 12 finals of the 2023 Women’s World Boxing Championships (New Delhi, India), organised by the International Boxing Association (IBA).

## 2. Materials and Methods

### 2.1. Experimental Design

An activity profile of the 12 final matches of the 2023 IBA Women’s World Boxing Championships was analysed according to the match phases of fighting (F), no-fighting (NF), and arbitral interruptions (AIs) (Table 1). During a match, the two boxers are constantly engaged in physical interaction, including punching, dodging, feinting, and parrying. These actions are not isolated but rather a dynamic exchange, where each boxer’s actions provoke and influence the other’s responses. For this reason, the fighting phase (Table 1) was considered as the interaction between the two boxers [14] and further divided into punching (P) and clinching (C) sub-phases [15]. According to the 2023 IBA rules, all matches were fixed in 3 rounds (1st, 2nd and 3rd) of 3 min, with 1 min of rest in between, for all 12 weight categories (i.e., 48 kg, 50 kg, 52 kg, 54 kg, 57 kg, 60 kg, 63 kg, 66 kg, 70 kg, 75 kg, 81 kg, and 81+ kg) [13].

### 2.2. Procedures

All match recordings were downloaded from the YouTube IBA Channel [16]. For this reason, since there is free public access, no informed consent was required according to the ethical standards outlined by the local research ethics committees [17].

The frequency of occurrence of all match phases was counted and annotated by means of a customised Excel dashboard (Microsoft Office 365, setup.exe version 16.0). The duration of each phase was measured using a free video annotation tool designed for motion analysis (Kinovea©, 0.9.5 release), with speed adjustable in 0.1 s, which allowed for slow-motion replay as well as rewinding the contest and watching events frame by frame. To further investigate the activity of each match, a specific ratio of F/NF total time was calculated for each weight category. A qualified researcher, who is a boxing coach certified by the Italian Boxing Federation [18], analysed footage of all matches and all rounds. The same researcher provided the analyses twice, four weeks apart, and subjected them to intra-observer reliability analysis (ICC = 1.0).

### 2.3. Statistical Analysis

This study provided descriptive statistics (frequency of occurrence, mean and standard deviation) for the count and duration of the match’s phases across all rounds (F, P, C, NF, and AI) for all weight categories. Since the study collected nominal data that were not normally distributed, a chi-square test was used to analyse the frequency of fighting phases across different weight categories and rounds, while a Kruskal–Wallis test was used to analyse the duration of each phase. Additionally, based on the research by Davis et al. [9], a Kruskal–Wallis analysis was performed, dividing the fighters into three weight groups: lightweight (LWC: 48 kg, 50 kg, 52 kg, 54 kg, 57 kg, and 60 kg); middleweight (MWC: 63 kg, 66 kg, 70 kg, and 75 kg); and heavyweight (HWC: 81 kg and over 81 kg).

The significance level was set at an alpha level of 0.05. Post hoc analyses for the chi-square test were performed using standardized residuals with a threshold of Z > 1.96, while for the Kruskal–Wallis test, the Bonferroni post hoc test was applied. All statistical analyses were performed using IBM SPSS Statistics ver. 25.0 (IBM Co., Armonk, NY, USA).

## 3. Results

Table 2 shows the counts and frequency of occurrence (%) of F, NF and AI phases in relation to rounds (first, second, third, and total ones) and weight categories. Collapsed data registered a total of 4108 counts (F + NF + AI) for the 12 weight categories.

The chi-square test showed a significant association between match phases and weight categories (χ^2^(22) = 155, *p* < 0.001). The post hoc test (standardized residuals) indicates no significant differences in the frequency of occurrence for the F phase, while significant differences for the AI and NF phases were found (Table 3).

The chi-square test between match phases (F, NF, AI) and rounds indicated no significant relationship (χ^2^(4) = 0.965, *p* = 0.915), while, when F was divided for P and C sub-phases, a significant association between weight categories (χ^2^(11) = 65.9, *p* < 0.001) was found (Table 4).

Finally, Table 5 shows the frequency of occurrence (%) of punching and clinching in the different rounds. The chi-square indicated no significant association (χ^2^(2) = 0.116, *p* = 0.944) in the distribution of P and C across the three rounds.

Table 6 shows the descriptive statistic (mean ± SD) of the duration of F phases (P and C), NF and AI in relation to each round for all weight categories.

The Kruskal–Wallis test showed significant differences for all phases in relation to the weight category: P (H_11_ = 74.563, *p* < 0.001), C (H_11_ = 29.782, *p* = 0.002), NF (H_11_ = 108.978, *p* < 0.001) and AI (H_11_ = 32.032 *p* < 0.001). The post hoc analysis revealed that the 63 kg weight category registered a higher number of significant differences with other categories (Table 7).

Considering the P phase, the Kruskal–Wallis test showed significant differences in all three rounds (first round: H_11_ = 46.048, *p* < 0.001; second round H_11_ = 47.086, *p* < 0.000; third round: H_11_ = 31.976, *p* < 0.001). The Bonferroni pairwise comparison showed differences between the 54 kg and 48 kg categories (*p* = 0.006), 54 kg and 57 kg (*p* = 0.005), 75 kg and 81+ kg (*p* = 0.008) in the first round; between the 70 kg and 52 kg (*p* = 0.036), 75 kg and 48 kg (*p* = 0.029), 75 kg and 52 kg (*p* < 0.001), 75 kg and 57 kg (*p* = 0.004), 75 kg and 63 kg (*p* = 0.001) in the second round; and, finally, between 50 kg and 52 kg (*p* = 0.040) in the third round.

For the C phase, the analysis indicated only a significant difference in the third round (H_11_ = 25.032, *p* = 0.009), while the Bonferroni comparison did not maintain the significant difference.

Regarding the ratio between the total time spent in fighting and no-fighting phases (F/NF), Table 8 shows that in the 54 kg, 57 kg, 60 kg, and 75 kg weight categories, there is more time spent in the NF phase than the F phase for all three rounds. In the 48 kg, 50 kg, 63 kg, and 81+ kg weight categories, more time was recorded in the F phase compared to the NF phase. Finally, for the 52 kg, 66 kg, 70 kg, and 81 kg weight categories, there was no prevalence in terms of time spent between the two phases among the rounds.

Considering the analysis in relation to the three weight divisions (LWC, MWC, HWC) and phases, the Kruskal–Wallis test identified a significant difference in the duration only for the F phase (H2 = 16.794, *p* < 0.001). In particular, the pairwise comparison showed a main effect between MWC and LWC (*p* = 0.007) and between MWC and HWC (*p* < 0.001). Considering the analysis between the three weight divisions and F phases (P and C) between the three rounds, the Kruskal–Wallis analysis showed significant differences only for P (H2 = 12.689 *p* = 0.002) in the first round. The Bonferroni post hoc test indicated main effects between LWC and HWC (*p* = 0.002; LWC = 1.56 ± 1.07, HWC = 1.96 ± 1.25) and between MWC and HWC (*p* = 0.002; MWC = 1.47 ± 0.85, HWC = 1.96 ± 1.25).

## 4. Discussion

As far as we know, this study is the first attempt to provide a complete picture of the time–motion analysis of the Women’s World Boxing Championship in the 3 × 3-min match format. This study investigates how often and how long the two boxers interact during the fighting (F) and no-fighting (NF) phases, as well as how often and how long there are arbitral interruption phases, across all rounds, weight categories, and weight division matches.

The main findings of the analysis of match phases’ counts and frequency of occurrence revealed the following: (a) As expected, F consistently displayed higher values compared to both NF and AI phases across all rounds and weight categories. (b) A significant association between match phases and weight categories was found; however, the standardised residual analysis showed significant associations (over-represented or under-represented) for NF and AI phases for eight weight categories (48 kg, 50 kg, 54 kg, 63 kg, 66 kg, and 70 kg). (c) No significant associations between rounds within the F, NF, and AI phases were found. (d) A significant relationship between weight categories and the two sub-phases of fighting (i.e., P and C) was observed, while their distribution did not vary significantly between rounds.

Furthermore, the main findings of the analysis of the durations revealed that: (e) with pooled data for the three rounds, the mean duration of the single P, C, NF, and AI phases showed significant differences among the weight categories; (f) in relation to P and C sub-phases, a significant difference was found in the mean duration of the single actions across all rounds for P, while no differences were found for C; (g) the analysis of the F/NF total time ratio for each weight category showed a prevalence in all rounds of F over NF for 48 kg, 50 kg, 63 kg, and 81+ kg weight categories, while the 54 kg, 57 kg, 60 kg, and 75 kg categories were characterised by a prevalence of NF over F; (h) for the weight divisions, the mean time spent in F and NF phases was significantly shorter in all rounds for MWC compared to HWC and LWC; (i) the mean duration of the P actions was significantly longer in the first round for HWC compared to LWC and MWC divisions.

As anticipated, the F phases accounted for the majority of the actions performed by the two boxers during a match, compared to NF and AI ones. This finding reflects the fundamental nature of Olympic boxing, where the goal is to score punches while maintaining high levels of activity, resulting in more time spent on active interactions (i.e., technical exchanges). This aspect also suggests that all female boxers in these finals’ world championship were able to sustain a high frequency of fighting activity throughout the three rounds, highlighting their very good level of physical preparation. However, although less frequent and with shorter durations, the NF (in particular) and AI phases contribute to determining the intermittent nature of boxing performance. Indeed, NF, which includes moments of lower engagement and tactical pauses, and AI naturally contribute to the overall time structure of the match and, together with the brief length of bouts, contribute to the reduced performance decline across rounds, which can be observed in amateur boxing [19]. The significant association found between weight categories and the frequency of occurrence of the NF and AI phases indicates that female boxers differently manage the match’s pacing and flow depending on their weight category. In fact, while the frequency of F phases remained stable across categories, the variation in NF phases suggests that the boxers may adopt distinct tactical approaches in the different weight categories. Regarding AI, the two lightest-weight categories (48 and 50 kg) and the two heaviest ones (81 and 81+ kg) were over-represented, as shown in Table 3. Even if these hypotheses warrant further investigation, they suggest that the nature of arbitral interruptions may reflect different strategic and tactical profiles depending on the weight category. Specifically, for the lightest categories, the over-representation of AI phases might be linked to more rapid exchanges or movement-based infractions, such as head clashes or foot fouls, which are more likely to occur due to faster pacing and greater mobility. On the other hand, for the heaviest categories, the higher number of AI phases might be related to less movement and more clinching or holding as fatigue increases, especially in later rounds. Additionally, referees may apply stricter control due to the higher impact of punches in these categories, leading to more frequent interventions. Furthermore, the absence of significant differences in the frequency of occurrences of the F, NF, and AI phases across the three rounds suggests that female boxers maintained a consistent fighting pattern throughout the match. This steady pattern shows that the boxers are well prepared physically and compete at a high level, as mentioned in an earlier review on amateur boxing [20], which found that the rates of technical actions varied greatly between different weight classes in beginner and elite boxers.

In higher international competitions, the tactical behaviour and fighting style of boxers might reflect the boxing cultures of the national school of boxers, even about the continent of origin. For example, the Brazilian female fighting style can be recognised for its defensive technique and very effective footwork and timing (as the Cuban school is, too). Unlike the action of Eastern European boxers, which is more oriented to an offensive behaviour, the Chinese style is based on a tactical approach that prioritizes precision over power. So, although not directly investigated in this study, this could demonstrate how tactical behaviour may vary significantly based on boxing culture and geographic origin.

In terms of the mean number (counts) of actions per round, the results of our study differ from the literature on elite boxers, like a study by Siska et al. [21], which found a value of 47.3 ± 3.8 active actions per round. Such a difference between the two studies could be attributed to two main factors. First, the study of Siska et al. [21] investigated male boxers from only three matches (64, 69 and 75 kg weight categories). Second, in their study, the active phases were limited to only P actions, while the C ones were not included. The main explanation of this is that Siska et al. [21] considered the C as a phase in which the two boxers give up fighting by hooking to block any kind of attack from the opponent. On the contrary, in our study, C was included in F since most of these actions were characterised by strikes exchanged between the two female boxers, with referees intervening only in cases of prolonged clinch time. In this regard, with data pooled for weight categories, our study found a C phase mean duration of 4.06 ± 2.27 s. These findings are in contrast with those from similar research conducted by Davis et al. [12] on elite female boxers using a 4 × 2-min match format (i.e., the official format during the 2012 Olympic Games) that showed how clinch time increased as the number of rounds increased. This discrepancy may be attributed to differences inherent in the 3 × 3-min match format (i.e., the official format during the 2023 World Championship), which may reflect a drastic change in the fighting strategy [22] of elite female boxers over time due to regulatory changes or a higher technical level and physical preparation of the competitors. In fact, the 3 × 3-min format, which became the standard format for the Olympic Games, appears to promote a more continuous fighting style, minimizing fragmentation and allowing the clinch to be utilized as an active component of the bout rather than simply an interruption.

The significant association between weight categories and P and C suggests that female boxers from different weight categories adopted distinct tactical approaches. For example, as shown in Table 4, boxers in the 54 kg and 57 kg categories mostly used punching actions, with P frequencies > 90% across all rounds (e.g., 98.2% of P in the first round) and a reduced use of the C actions. In contrast, heavier categories such as 81 kg and 81+ kg showed a progressive increase in the use of C actions (e.g., reaching up to 22.9% in the third round for 81+ kg). These data clearly reflect distinct tactical approaches based on boxers’ weight categories, highlighting the need for weight-specific training strategies that reflect each weight category’s typical fighting patterns and technical–tactical requirements.

Although no statistically significant differences were found in the distribution of P and C phases between rounds, a visual analysis of Table 4 reveals notable trends in some weight categories. Specifically, the 48 kg, 50 kg, 60 kg, and 75 kg weight categories showed an increase in the frequency of occurrence of P phases from the first to the second round, followed by a decrease from the second to the third round. These categories appear to align with the “inverted U profile” observed in elite male boxers competing in the 3 × 3-min format [9]. Conversely, the 52 kg and 70 kg categories exhibited a consistent increase in the frequency of occurrence in both the P and C phases from the first to the third round. This pattern resembles what Davis et al. [1] identified as “profile J” in their study on novice male boxers using a 3 × 2-min format. Lastly, the 81 kg and +81 kg categories displayed a kind of “declining profile”, with higher P frequency during the first round that gradually decreased in the second and third rounds. All these discrepancies might be attributed to the specific characteristics of the 3 × 3-min format.

The analysis of the mean duration of single P, C, NF actions and AI phases with pooled data for rounds revealed significant differences between weight categories. For instance, in the 50 kg category, the average duration of the P phase was 1.48 ± 0.35 s, compared to 1.74 ± 0.32 s in the 81+ kg category. Similarly, the C phase showed more significant variability across weight categories. For example, in the 60 kg category, the average C phases duration reached 4.98 ± 0.60 s, while in the 54 kg category, it was considerably shorter, with an average of 2.29 ± 0.86 s. These results emphasize the value of providing detailed time–motion analysis and support the need for training programs tailored to the specific demands of each weight division. According to Gutiérrez-Santiago et al. [23], the duration of NF and AI phases significantly contributes to the intermittent structure of combat sports and influences the overall energy demands and match dynamics. In this regard, our results showed relatively longer NF phase durations (4.65 ± 3.14 s) in the 60 kg compared to shorter durations (1.84 ± 1.32 s) in the 63 kg category. Regarding AI, in the 81+ kg category, the mean duration was 4.60 ± 1.57 s, while in the 50 kg category, it was slightly longer, averaging 6.38 ± 2.54 s.

Usually, the duration of a round might be influenced by referees’ stoppages and knockouts or TKOs, which are regulated by a stop of the official time, either increasing or decreasing the round duration, while referees’ breaks are not regulated by a stop of the time. Therefore, a correct approach of the analysis should consider it. According to the literature, the only study examining time–motion analysis in women’s boxing was conducted by Davis et al. [12]. The authors found an average round duration of 132 s, with an activity-to-break ratio between rounds of 6.6:1. Additionally, this study noted that the frequency and duration of the referee’s stoppage increased from the first to the fourth round. However, when we compare these results with those of our study, we identify two main differences. First, it is important to note that the study of Davis et al. [12] was conducted during the London 2012 Olympics when the fight format consisted of 4 × 2-min rounds. Second, in all rounds and all finals in our study, the referees never assigned stoppages, knockouts or TKOs. For this reason, in our study, the duration of all rounds was always 180 s. This fact suggests that there might be differences in refereeing management and combat dynamics in different competitive contexts. Nevertheless, more research is required to fully comprehend how refereeing factors affect match length and, in turn, the physical and psychological demands required of female elite boxers.

While the duration of C actions across the three rounds remained relatively stable, the duration of P actions showed significant differences (Table 4). For example, in the 81+ kg category, the time taken for P actions gradually decreased, going from 2.04 ± 0.84 s in the first round to 1.76 ± 1.11 s in the second one and then to 1.41 ± 1.05 s in the third one, which might indicate tiredness or a way to save energy. We also observed a similar trend in the 60 kg weight category, where the P durations decreased from the first round. Conversely, in the 52 kg weight category, the P duration increased from 1.77 ± 1.01 s in the first round to 2.57 ± 1.69 in the second one before slightly decreasing to 2.00 ± 1.38 s in the third round. These findings confirm that, unlike the C actions, the mean duration of P actions tends to vary significantly through the weight categories. This variability may be influenced by boxers’ fatigue, tactical adjustments, or their ability to maintain a high intensity of actions as the bout progresses. These facts indicate that, in elite female boxers, like elite male ones, sustaining such a high work rate did not adversely affect the boxers’ capacity to maintain match discipline. This finding also highlights the need for targeted training to sustain striking effectiveness throughout all rounds of high-level competition.

In general, boxers use the NF phase to study their opponents, prepare more precise attacks, move more around the ring, control directions and pace changes, and conserve energy during crucial moments of the match. In our study, the analysis of the F and NF phases in relation to the weight categories revealed the dominance of the NF phase in the 54 kg, 57 kg, 60 kg, and 75 kg categories. Indeed, these are the female Olympic categories [24]; this more methodical and reflective approach may be a response to the demands of high-level competitions, such as the Olympics, where every action must be well calculated to maximise the chances of success. This finding is in line with the results of a study on elite male boxers from the welter category (64–89 kg) in the London 2012 Olympics [25], which indicated that the percentage of time spent on motor actions (i.e., the effective time of travel with the lower body) represented more than half of the round time. On the contrary, in the 48 kg, 50 kg, 63 kg, and 81+ kg categories, a greater prevalence of the fighting phase over the NF one was observed. This fact suggests that the female boxers in these weight categories tended to adopt more continuous and direct strategies in their matches.

Furthermore, in the categories with a predominance of NF, coaches could focus on decision-making drills, ring control exercises, and tactical sparring, aiming to develop the ability to read the opponent and choose the right moment to land. Conversely, in categories with a predominance of F phases, coaches may consider incorporating high-intensity interval training (HIIT) that mimics the typical duration of punching (P) phases, with repeated sprint and reaction drills, to sustain frequent and explosive exchanges.

In turn, these findings could also help implement injury prevention strategies. In fact, despite the injuries, the risk of injuries in amateur boxing is lower than in other full-contact combat sports, such as taekwondo and karate, and that injuries in training and in competition differ significantly in characteristics; in boxing, the injury incidence is higher in competition than in training [26,27]. For instance, punches involve high-speed muscular actions, requiring muscle strength transmission through the boxer’s entire kinetic chain, starting with lower body ground and transferring energy to upper extremities. So, knowing the frequency and duration of the match’s technical exchanges may assist coaches in customising training programs that mimic the structure of the boxers’ interaction, reducing the injury risks and promoting the long-term health of their athletes.

We also conducted the same analysis with data divided into three weight divisions (LWC, MWC, and HWC), for which the data showed significant differences among them for the F and NF phases. Specifically, the mean duration of F actions was 2.2 ± 1.9, 1.9 ± 1.5 and 2.4 ± 1.9 s for LWC, MWC and HWC weight divisions, respectively. These results indicate that the female boxers in MWC categories displayed shorter action durations for F, as well as for NF, than their LWC and HWC counterparts. Furthermore, the P sub-phase registered significant differences for rounds, particularly in the first round. The HWC division showed the longest duration (1.96 ± 1.25 s) compared to the other two divisions (LWC: 1.56 ± 1.07 s; MWC: 1.47 ± 0.85 s). This increased the mean duration of the interaction.

These findings have important implications for training methodology. For instance, heavier weight categories showed more significant reliance on the C phases and decreased P phase frequency across rounds, suggesting a need to improve continuity and offensive output through targeted conditioning and tactical drills. Conversely, lighter categories, which favoured shorter and more frequent P phases, may benefit from training programs that emphasize speed, reactivity, and rapid transitions. Furthermore, the observed variability in the duration of AI and NF phases between categories confirms the importance of including simulated interruptions and tactical breaks in training scenarios.

Understanding the phases and characteristics of each weight category could assist coaches in designing training programs that are specifically tailored to meet the needs of each athlete based on their weight category. This approach could reduce the reliance on extreme and potentially harmful weight cuts, promoting the health of female boxers and enhancing their performance. Furthermore, the need for targeted training extends beyond technique alone and includes healthy weight management, which can significantly impact performance. Studies have shown that extreme weight management practices, such as dehydration and restrictive dieting, can have long-term detrimental effects on the physical and psychological health of female boxers, impairing their performance and leading to eating disorders and body dissatisfaction, even after retirement [28].

A limitation of this study is the small sample size, which only includes the 12 weight category finals of the 2023 Women’s Boxing World Championships, which could potentially affect the generalisability of our findings to a wider female boxing population competing at different levels (e.g., regional and national competitions). However, this study aimed to analyse the highest level of female boxing competitions by focusing exclusively on the final matches. Therefore, the decision to analyse only these finals was based on the assumption that they reflect the conditional, technical, and tactical levels of boxers’ performances during a world championship. Indeed, we expect the intensity and quality of performance to remain high throughout the entire tournament, despite potential tactical adjustments. Additionally, while the time–motion analysis provides valuable insights into the activity profiles of elite female boxers, it does not consider factors such as fatigue, psychological aspects, or external influences like environmental conditions.

Finally, future studies analysing the most competitive tournaments (i.e., World Championships and Olympic Games) will benefit from including all matches to provide a more comprehensive understanding of performance patterns throughout the entire tournament. Moreover, this kind of analysis can be further extended to investigate the impact of fatigue and recovery, as well as psychological factors, at lower competitive levels (e.g., regional and national).

## 5. Conclusions

In conclusion, this study comprehensively analysed the temporal dynamics and activity phases in elite women’s Olympic boxing, focusing on the 3 × 3-min fight format. The findings offer valuable insights for designing specific training programs, indicating that training strategies should be tailored according to weight category and the number of rounds and in managing interactions between the two boxers during the fighting phases. By analysing the frequency and duration of fighting (F), no-fighting (NF), and arbitral interruption (AI) phases across all rounds and weight categories, we observed significant differences in how athletes manage match dynamics depending on their weight category. While the F phase dominated the activity structure, the variations in NF and AI phases highlighted distinct tactical approaches.

Finally, it is interesting to note that some of the currently recognized Olympic weight categories, such as 54 kg, 57 kg, 60 kg, and 75 kg, showed a higher incidence of NF phases. This may reflect a more strategic approach to match management, characterized by deliberate rhythm, determined pauses, and greater attention to opponent analysis. Such behaviour, likely influenced by experience in high-level international competitions, could serve as a helpful model for non-Olympic categories, which may benefit from incorporating similar tactical strategies into their technical preparation.

These findings suggest that a successful training program that replicates the temporal structure of matches, particularly the alternation between short bursts of punches at long range and in close range (C), interspersed by tactical pauses (NF), should also be tailored to the predominance of these phases in certain weight categories. Additionally, for heavier weight categories, where C actions are more frequent and prolonged, coaches should include isometric hold exercises and drills to improve close-range techniques, while for lighter categories, they should focus on enhancing agility and decision making.

As the participation of women in boxing continues to grow at the international level, there is a pressing need to expand research focused on women’s boxing. Despite the significant increase in female competitors, studies on this topic remain relatively scarce. Therefore, this analysis represents an important first step toward a better understanding of elite women’s boxing while also underscoring the necessity for further research to support its development. Future research could provide additional insights into optimizing performance in women’s boxing, for instance, tailoring training interventions that consider the boxers’ recovery strategies, to which it is also necessary to investigate the coaching staff level of knowledge on this specific issue [29].

## Figures and Tables

**Table 1 sports-13-00187-t001:** Classification of each phase and fighting phase.

Phase	Description	Transition Criteria
Fighting (F)	begins when the two boxers leave the safety distance (i.e., the neutral distance where neither boxer can land an effective punch, even with a fully extended lead arm, making direct attacks and physical contact impossible) to move to a fighting distance and begin the technical exchange	From NF to F: Occurs when a boxer reduces the distance, moving out of the safety zone. This transition was observed when a boxer intentionally closed the distance with the purpose of initiating an attack.
Punching (P)	i.e., technical exchanges of punches	
Clinching (C)	i.e., when one or both boxers engage in close-range combat, resulting in a momentary locking of their arms	
No-fighting (NF)	when the two boxers finish the technical exchange and position themselves at a safe distance, maintaining a study attitude (i.e., tactical periods in which athletes, maintaining a safe distance, prepare for an attack by adopting either fighting or no-fighting stances)	From F to NF: occurred when a boxer, after an attack or defensive move, used a leg movement (e.g., retreating) to create a distance where neither boxer could immediately strike. This was observed through their movement and positioning.
Arbitral Interruption (AI)	i.e., during prolonged holding or fouls	From F or NF to AI: when the referee made a visible gesture, usually a hand signal indicating ‘break’. The phase began as soon as the referee initiated the hand signal, regardless of whether the boxers had been disengaged or were still in contact. It ended when the referee issued the verbal or visual command “boxe,” signaling the official resumption of the bout. This transition is clearly defined by the referee’s actions and is objective in nature.

**Table 2 sports-13-00187-t002:** Count and frequency of occurrence (%) of these changes in activities in relation to the different match phases (F, NF and AI) and rounds (1st, 2nd and 3rd) for each weight category.

	Fighting			No-Fighting			Arbitral Interruption			Total Activities (F + NF + AI)
	**Round**			Round			Round			
weight categories	1st n (%)	2nd n (%)	3rd n (%)	1st n (%)	2nd n (%)	3rd n (%)	1st n (%)	2nd n (%)	3rd n (%)	
48 kg	81 (75.7)	85 (76.6)	90 (72.6)	14 (13.1)	16 (14.4)	23 (18.5)	12 (11.2)	10 (9.0)	11 (8.9)	342
50 kg	38 (67.9)	50 (65.8)	45 (66.2)	10 (17.9)	17 (22.4)	15 (22.1)	8 (14.3)	9 (11.8)	8 (11.8)	200
52 kg	81 (73.6)	76 (73.1)	62 (66.0)	21 (19.1)	24 (23.1)	28 (29.8)	8 (7.3)	4 (3.8)	4 (4.3)	308
54 kg	118 (69.4)	103 (75.2)	111 (74.0)	52 (30.6)	32 (23.4)	38 (25.3)	0	2 (1.5)	1 (0.7)	457
57 kg	55 (69.6)	46 (71.9)	61 (67.0)	22 (27.8)	17 (26.6)	27 (29.7)	2 (2.5)	1 (1.6)	3 (3.3)	234
60 kg	70 (73.7)	67 (72.0)	69 (67.6)	22 (23.2)	24 (25.8)	28 (27.5)	3 (3.2)	2 (2.2)	5 (4.9)	290
63 kg	130 (73.4)	105 (75.5)	119 (75.8)	46 (26.0)	32 (23.0)	36 (22.9)	1 (0.6)	2 (1.4)	2 (1.3)	473
66 kg	81 (73.6)	77 (75.5)	89 (71.8)	27 (24.5)	23 (22.5)	34 (27.4)	2 (1.8)	2 (2.0)	1 (0.8)	336
70 kg	106 (75.2)	103 (68.2)	112 (70.0)	34 (24.1)	47 (31.1)	47 (29.4)	1 (0.7)	1 (0.7)	1 (0.6)	452
75 kg	99 (71.7)	87 (69.6)	99 (73.3)	32 (23.2)	32 (25.6)	30 (22.2)	7 (5.1)	6 (4.8)	6 (4.4)	398
81 kg	67 (72.8)	52 (66.7)	51 (63.0)	19 (20.7)	18 (23.1)	19 (23.5)	6 (6.5)	8 (10.3)	11 (13.6)	251
81+ kg	78 (72.9)	86 (71.7)	105 (75.0	21 (19.6)	24 (20.0)	25 (17.9)	8 (7.5)	10 (8.3)	10 (7.1)	367

Note. F = fighting phase; AI = arbitral interruption phase; NF = no-fighting phase.

**Table 3 sports-13-00187-t003:** Post hoc test (standardized residual) for F, AI and NF in relation to weight categories.

		Match Phase		
		F	AI	NF
weight category	48 kg	1.27	5.04 *	−3.75 #
	50 kg	−1.74	5.81 *	−0.94
	52 kg	−0.36	0.78	0.00
	54 kg	0.38	−4.10 #	1.56
	57 kg	−0.93	−1.37	1.64
	60 kg	−0.34	−0.77	0.73
	63 kg	1.52	−3.72 #	0.18
	66 kg	0.69	−2.67 #	0.55
	70 kg	−0.44	−4.06 #	2.41 *
	75 kg	−0.13	0.45	−0.08
	81 kg	−1.52	4.52 *	−0.56
	81+ kg	0.59	3.26 *	−2.19 #

* Overrepresented; # Underrepresented. Note. F = fighting phase; AI = arbitral interruption phase; NF = no-fighting phase.

**Table 4 sports-13-00187-t004:** Post hoc test (standardized residual) for P and C in relation to weight categories.

		Fighting Phases	
		P	C
weight category	48 kg	−4.21 #	4.21 *
	50 kg	−3.73 #	3.73 *
	52 kg	−1.59 #	1.59 *
	54 kg	3.47 *	−3.47 #
	57 kg	3.15 *	−3.15 #
	60 kg	2.02 *	−2.02 #
	63 kg	0.9	−0.9
	66 kg	−0.377 #	0.377 *
	70 kg	2.2	−2.2
	75 kg	−0.042	0.042

* Overrepresented; # Underrepresented. Note. P = punching; C = clinching.

**Table 5 sports-13-00187-t005:** Frequency of occurrence (%) of punching and clinching in different rounds.

		Rounds					
		1st		2nd		3rd	
		Punching (%)	Clinching (%)	Punching (%)	Clinching (%)	Punching (%)	Clinching (%)
weight category	48 kg	79	21	81	19	78.7	21.3
	50 kg	73.7	26.3	82	18	75.6	24.4
	52 kg	76.5	23.5	86.8	13.2	91.9	8.1
	54 kg	97.5	2.5	90.3	9.7	92.8	7.2
	57 kg	98.2	1.8	93.5	6.5	95.1	4.9
	60 kg	91.4	8.6	92.5	7.5	92.8	7.2
	63 kg	82.2	10.8	90.5	9.5	88.2	11.8
	66 kg	90.1	9.9	80.5	19.5	89.9	10.1
	70 kg	80.2	19.8	96.1	3.9	98.2	1.8
	75 kg	86.9	13.1	88.5	11.5	87.9	12.1
	81 kg	89.6	10.4	86.5	13.5	80.4	19.6
	81+ kg	92.3	7.7	82.6	17.4	77.1	22.9

**Table 6 sports-13-00187-t006:** Duration (mean ± SD) of match phases.

		Weight Category											
		48 kg	50 kg	52 kg	54 kg	57 kg	60 kg	63 kg	66 kg	70 kg	75 kg	81 kg	81+ kg
1st round	P	2.60 ± 1.9	1.77 ± 1.5	1.77 ± 1.0	1.14 ± 0.5	1.77 ± 0.7	1.34 ± 0.7	1.47 ± 0.7	1.79 ± 1.4	1.44 ± 0.7	1.26 ± 0.6	1.88 ± 1.6	2.04 ± 0.8
	C	3.47 ± 2.1	4.88 ± 1.9	5.28 ± 5.1	1.66 ± 0.9	3.33	5.38 ± 1.0	3.47 ± 1.4	4.63 ± 1.2	3.82 ± 1.1	3.10 ± 2.5	4.01 ± 1.7	4.06 ± 2.5
	NF	2.98 ± 1.7	4.02 ± 3.1	2.40 ± 1.8	2.26 ± 1.6	6.09 ± 6.2	5.21 ± 2.4	1.71 ± 1.1	3.61 ± 2.6	2.50 ± 2.2	2.76 ± 1.3	5.16 ± 3.6	3 ± 1.9
	AI	5.52 ± 3.8	6.01 ± 1.8	5.82 ± 2.2	/	4.75 ± 1.7	2.72 ± 1.2	5.53	5.22 ± 0.1	4.47	3.20 ± 1.3	5.85 ± 1.9	5.19 ± 1.1
2nd round	P	2 ± 1.4	1.58 ± 1.0	2.57 ± 1.7	1.65 ± 1.3	2.35 ± 2.0	1.10 ± 0.5	1.87 ± 1.2	1.69 ± 1.1	1.18 ± 0.4	1.10 ± 0.4	1.33 ± 0.8	1.76 ± 1.1
	C	4.52 ± 2.2	4.73 ± 3.0	4.69 ± 2.7	3.27 ± 1.4	5.40 ± 6.1	5.26 ± 2.1	3.95 ± 1.8	4.67 ± 1.8	2.87 ± 1.2	4.69 ± 2.2	5.12 ± 2.7	3.68 ± 2.3
	NF	2.97 ± 2.1	3.69 ± 1.7	3.53 ± 2.1	3.05 ± 2	7.08 ± 5.4	5.13 ± 3.6	2.16 ± 1.7	3.53 ± 2.0	2.36 ± 1.7	2.78 ± 2.0	4.80 ± 3.1	1.95 ± 1.7
	AI	4.62 ± 2.3	5.83 ± 2.4	3.71 ± 1.7	6.67 ± 1.0	4.03	2.95 ± 1.2	9.30 ± 1.5	4.85 ± 0.1	3.73	3.69 ± 0.8	4.90 ± 1.5	5.91 ± 3.8
3rd round	P	1.3 ± 0.8	1.10 ± 0.6	2 ± 1.38	1.55 ± 1.0	1.42 ± 0.9	1.20 ± 0.6	2.04 ± 1.4	1.60 ± 1.0	1.17 ± 0.3	1.28 ± 0.8	1.50 ± 1.1	1.41 ± 1.1
	C	2.95 ± 1.5	5.07 ± 0.7	4.24 ± 3.5	1.94 ± 0.6	2.38 ± 1.0	4.29 ± 1.0	5.28 ± 2.0	3.21 ± 1.7	2.07 ± 0.1	3 ± 1.8	4.40 ± 1.4	4.13 ± 3.1
	NF	2.78 ± 2.3	3.73 ± 2.2	3.57 ± 2.8	2.80 ± 1.7	4.48 ± 4.1	3.62 ± 3.4	1.64 ± 1.2	2.61 ± 1.6	2.36 ± 1.3	3 ± 1.9	2.81 ± 2	2.01 ± 1.4
	AI	4.13 ± 2.6	7.31 ± 3.4	3.78 ± 1.5	4.57	6.87 ± 5.6	3.95 ± 1.3	6.60 ± 0.6	8.7	11.6	4.80 ± 3.1	5.04 ± 2.2	3.77 ± 1.6

Note. P = punching; C = clinching; AI = arbitral interruption phase; NF = no-fighting phase.

**Table 7 sports-13-00187-t007:** The Bonferroni pairwise comparison for all phases in relation to the weight category.

			Weight Categories											
			48 kg	50 kg	52 kg	54 kg	57 kg	60 kg	63 kg	66 kg	70 kg	75 kg	81 kg	81+ kg
weight categories	48 kg	P												
		C												
		NF												
		AI												
	50 kg	P			*									
		C												
		NF												
		AI												
	52 kg	P												
		C												
		NF												
		AI												
	54 kg	P			*				*					
		C		**				**						
		NF					***	***				***		
		AI												
	57 kg	P												
		C												
		NF												
		AI												
	60 kg	P												
		C												
		NF												
		AI		****					****					
	63 kg	P												
		C												
		NF	***	***	***	***	***	***		***		***	***	
		AI												
	66 kg	P												
		C												
		NF												
		AI												
	70 kg	P			*				*					
		C												
		NF		***				***					***	
		AI												
	75 kg	P	*		*		*		*	*				*
		C												
		NF												
		AI		****										
	81 kg	P												
		C												
		NF												
		AI												
	81+ kg	P												
		C												
		NF		***			***						***	
		AI												

Note. * P = punching; ** C = clinching; *** NF = no-fighting phase; **** AI = arbitral interruption phase.

**Table 8 sports-13-00187-t008:** Ratio between the total time spent in the fighting and no-fighting phases for each weight category.

		1st Round		2nd Round		3rd Round	
		F > NF	NF > F	F > NF	NF > F	F > NF	NF > F
weight categories	47 kg	1.85		1.99		1.20	
	50 kg	1.45		1.11		1.02	
	52 kg	1.75		1.18			0.77
	54 kg		0.56		0.76		0.65
	57 kg		0.29		0.45		0.34
	60 kg		0.54		0.42		0.52
	63 kg	1.16		1.23		1.97	
	66 kg		0.86	1.23			0.94
	70 kg	1.19			0.59		0.55
	75 kg		0.80		0.80		0.71
	81 kg		0.67		0.69	1.35	
	81+ kg	1.13		1.93		1.72	

Note. F = fighting phase; NF = no-fighting phase.

## Data Availability

Data are contained within the article.

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
