# Peer review of "Time–Motion Analysis of the 2023 Women’s World Boxing Championships Finals"

_sports, 2025, doi:10.3390/sports13060187_

Round 1
Reviewer 1 Report
Comments and Suggestions for Authors
The paper should be supplemented with limitations and future research directions. Although the structure of women's boxing is well established and explained in the paper, it should be recognized that only the final fights are included in the paper. In this way, we do not have an insight into the entire structure of women's boxing. It is very likely that the technical-tactical performance is different in the fights until the final.
In the 495th line, the author Milivoj, D. is listed. It should be changed to Dopsaj, M. Dopsaj is his last name, and Milivoj is his first name.
Author Response
Dear Reviewer #1
We deeply thank you for your valuable suggestions and comments on our manuscript. Those comments are of enormous assistance to us for improving and revising our manuscript. We have studied the comments carefully and made corrections in line with the suggestions you made.
In attachment you can see our answers

Reviewer 2 Report
Comments and Suggestions for Authors
First of all, thank you very much for allowing me to review this manuscript. I congratulate the authors for the work done. Some suggestions for improvement are:
Abstract
Include clearly the objective of the manuscript.
In the results section, please include the p values.
Introduction
Congratulations, it is very well elaborated and well understood.
It would be interesting to include a hypothesis at the end of the introduction.
Material and methods
It is well developed and explained.
Is it sufficient and representative to analyze only a final figth for each of the categories? Previous studies have done it in a similar way? I am mainly referring to consider whether the sample size of videos analyzed is sufficient to draw conclusions. Please clarify this question.
I would define the first part of the method with a subsection (design, procedure, or similar). Therefore, the statistical analysis would no longer be 2.1
. Did only one researcher analyze the videos? Or was an analysis performed by two researchers who subsequently tried to find agreement on what happened during the combat?
Results
This information “The Chi-square test between match phases (F, NF, AI) and rounds did not indicate a 159 sig- nificant relationship (χ² (4) = 0.965, p = 0.915), while when F was divided by P and C 160 subphases a significant association was found between weight categories (χ² (11) = 65.9, p < 0.001) 161” is represented in any table?
I recommend that you place Table 5 in horizontal layout to facilitate understanding.
Discussion
Very well developed. However, a paragraph of limitations of the study is missing.
Conclusion
Responds correctly to the objective of the study.
Author Response
Dear Reviewer #2
We deeply thank you for your valuable suggestions and comments on our manuscript. Those comments are of enormous assistance to us for improving and revising our manuscript. We have studied the comments carefully and made corrections in line with the suggestions you made.
In attachment you can see our answers

Reviewer 3 Report
Comments and Suggestions for Authors
Dear Authors,
I have received your manuscript, "Time-Motion analysis study of the 2023 Women's World Boxing Championships Finals". This study presents a comprehensive and methodologically rigorous time-motion analysis of female boxing under the 3x3 minute format. The manuscript's structure is logical, and statistical analyses of fight dynamics across weight categories and rounds. The results offer valuable insights into the intermittent nature of female boxing and its tactical components. Nevertheless, several aspects would benefit from clarification or revision before consideration for publication.
Abstract:
The abstract is well-organized and outlines the key findings of the study. However, some clarifications could enhance its precision and accessibility:
- Explicitly state the total sample size.
- Define all acronyms at first use (e.g., LWC, MWC, HWC, AI).
- The conclusion would benefit from one practical implication (e.g., how findings could influence training design or weight-specific tactics).
Introduction:
The introduction provides a comprehensive context and justifies the need for a female-specific TMA in boxing. It traces the development of female boxing and underlines the scarcity of relevant studies post-2012. However, I recommend:
- Clearly define the difference between "Fighting" and its subcomponents (Punch and Clinch) earlier in the text.
- Expanding on the rationale for using the 3x3 minute format and its implications for physiological demands compared to previous formats (e.g., 4x2 minutes).
- Clarify if this study only includes winners or both boxers per final, as this affects generalizability.
Methodology:
The methodology is clearly described and replicable. The use of Kinovea for motion annotation and intra-observer reliability is a strength. Suggestions:
- Justify the final round selection of each weight category, as finals may differ tactically from earlier rounds.
- Describe how the observer determined transitions between F, NF, and AI phases. Were there objective thresholds (e.g., distance or time) or subjective judgment?
- Consider clarifying whether inter-observer reliability was assessed, even though intra-observer ICC was excellent.
Results:
There are present detailed and well-structured tables. The use of Chi-square, Kruskal-Wallis, and post-hoc analyses is appropriate. Still, a few issues merit attention:
- The results section is sometimes overloaded with numbers. I recommend highlighting key takeaways after each table (e.g., “54 kg showed the lowest F/NF ratio, indicating more strategic pauses”).
- Table formatting needs refinement for clarity and legibility (several tables lack consistent alignment and font scaling).
- Clarify whether frequencies are per round or cumulative, and consider normalizing data for bouts with differing counts of actions.
Discussion:
The discussion interprets the data meaningfully and compares findings to previous studies. The section benefits from consideration of the tactical behavior observed in specific weight classes. Recommendations:
- Offer deeper speculation on the overrepresentation of AI in HWC (e.g., refereeing styles, fatigue, or movement limitations).
- Reflect on how tactical behavior may differ by boxing culture or continent, especially given the international nature of the sample.
- Consider suggesting training interventions based on F/NF ratios and the durations of Clinch and Punch phases (e.g., interval types, decision-making drills).
Conclusion:
The conclusion restates the findings but could be more practically oriented:
- Add one or two concrete takeaways for coaches or sports scientists (e.g., training should simulate …).
Author Response
Dear Reviewer #3
We deeply thank you for your valuable suggestions and comments on our manuscript. Those comments are of enormous assistance to us for improving and revising our manuscript. We have studied the comments carefully and made corrections in line with the suggestions you made.
In attachment you can see our answers

Reviewer 4 Report
Comments and Suggestions for Authors
Line 112-114
In addition to the analysis for weight categories, a further analysis was conducted grouping these in three weight divisions [9]: light weight categories (LWC: 48 kg, 50 kg, 52 kg, 54 kg, 57 kg and 60 kg), middle weight categories (MWC: 63 kg, 66 kg, 70 kg and 75 kg) and heavy weight categories (HWC: 81 kg and 81+ kg).
Comment 1
The results in all tables show results by individual categories. Where are the data according to the weight divisions LWC, MWC and HWC?
Line 101
All matches were fixed in 3 rounds (1st, 2nd and 3rd) of 3 minutes, with one minute of rest in between.
Comment 2
This is fine if there are no stoppages in the fight such as knockouts or TKOs. Describe how you recorded this information, whether there were any such situations and how many.
Line 127-128
The analyses were provided twice, four weeks apart, by the same researcher, and subjected to intra-observer reliability analysis (ICC = 1.0).
Comment 3
The researcher has done a lot of work. Each fight is rated 2× for both competitors, that's 24 reviews times 3 rounds. He reviewed 72 rounds twice (test-retest) and measured the duration of each situation identically both times?
Line 177
Table 6 shows the Bonferroni pairwise comparison.
Line 180
Table 6. The Bonferroni pairwise comparison for all phases in relation to the weight category
Comment 4
Some information is repeated.
Comment 5, Sample
What is the sample in this study 24 female boxers or 12 fights or 12 categories or 3 weight divisions?
The sample should be explained better.
Although there is a lot of work here, this is an extremely small sample. One category is described by 1 fight, so displaying data by division would be much better in this case.
Comment 6, Varijables
Why is a clinch considered a fight? If there is a fight in a clinch, it's ok, but if they block each other, then it's a nofight.
And the referee's stoppage is part of the nofight time, Variable NF + AI = total nofight time inside round.
The logic of dividing into variables needs to be better explained.
Comment 7, Results
The description of what we see in the table is usually below the table, not above.
The tables show the results by category, not by division. In such studies, we see better insight and differences between categories between weight divisions. It is unlikely that a difference of 2-3 kg will affect the tactics or structure of the fight, a difference of 18 kg (LWC) could be felt in the structure of the fight.
Comment 8, The tables are not clear
Table 4. Frequency of occurrence (%) of punch and clinch to the different rounds.
Table 4, 1 round (48kg), 79% punch and 21% clinch
If this is 100%, what is the arbitral Interruption phase and No-Fighting phase?
Table 2. frequency of occurrence (%) of these changes of activities in relation to the different match’s 148 phases (F, NF and AI) and round (1st, 2nd and 3rd) for each weight-category.
Table 2, 1 round (48kg) fighting, result is 81 (75.7%)
Is 81 time spent in fighting measured in seconds or frequency of occurrence for fighting?
Comment 9, discusion and conclusion
All the women who perform at the world championship are elite athletes, and the finalists are the elite within the elite. It is necessary to carefully analyze the results, these are more model values than the structure of the fight in women boxing.
Final comment, with this data an interesting paper can be done. But it needs to be structured differently. It needs to be based on weight divisions and not categories.
Author Response
Dear Reviewer #4
We deeply thank you for your valuable suggestions and comments on our manuscript. Those comments are of enormous assistance to us for improving and revising our manuscript. We have studied the comments carefully and made corrections in line with the suggestions you made.
In attachment you can see our answers

Reviewer 5 Report
Comments and Suggestions for Authors
The abstract is well constructed but is completely in agreement with the main text of the manuscript. Maybe the analysis technology could be added in the "Methods" section. The notion of the Olympic category comes too late in the conclusion and should be introduced earlier in the text, in the “Background” or “Methods” section.
The « introduction » section is well constructed too. The paragraphs of sports and scientific background perfectly clarify the subject of this study; they are pertinent, in accordance with the subject, and the references are recent and well adapted to the context. On the other hand, the research question could be framed in a more scientific context and, most importantly, should indicate the added value of this study. Additionally, it might be important to formulate one or more hypotheses so that the study appears more scientific and less observational.
Concerning the « methods » paragraph, the global item selection process is clear. However, some clarifications are necessary. Indeed, the notion of « safety distance » remains too vague at this stage and needs to be specified. Does it refer to the distance from which a strike can be effectively delivered, that is the zone of immediate danger? Should this distance be assessed while taking her own guard or the opponent’s guard position into account, or independently of it? Clarifying this concept is essential to ensure a relevant analysis of offensive or defensive behaviors. Moreover, from an organizational standpoint, this section could be structured differently. Indeed, a paragraph titled « 2.1 » appears at the very end of the section, whereas concepts related to techniques and boxing categories, as well as measurement and calculations, are introduced earlier and could have been organized into sub-sections. Many thanks to better organize. The sub-paragraph « statistical analysis » is clear.
The « results » paragraph is well constructed, although dense. Why, in the legend of figure 2, the authors indicate « changes of activities » and not « interaction », as in the methods? Moreover, The figure 5 seems to be incomplete; certain data points are missing, for instance, values below 1.3±0.76 (48kg / 3rd round). Maybe the table 6 could be described in text, because of low number of significances.
The « discussion » paragraph is interesting but too long. The restatement of the results and their structured presentation in four main points for occurrence and in five points for durations are particularly well received. On the other hand, the inclusion of numerical values in the text makes it difficult to read. Moreover, and this is a major point, it seems problematic to discuss the results despite a lack of statistical significance, based solely on visual analysis. Profiles are defined, but no differences can be statistically confirmed; please limit the discussion to what can legitimately be discussed. Some elements appear and seem more or less disconnected from the results, the topic of injuries for instance. This could be interpreted as perspectives, but they are not supported by the actual findings. Nothing suggests a link to injury prevention, except for the statement that this work could be useful for coaches aiming to optimize performance and prevent injuries. The relevance of using an analysis based on three groups is also questionable. Do these additional data provide any meaningful insights? Other extrapolations, particularly at the end of the discussion, seem speculative.
The conclusions are consistent with the content of the paper, and maybe too much. Indeed, the two first paragraphs seem to be inappropriate because of redundancy with the rest of the paper. It seems that only the third paragraph sounds like a conclusion and could serve as a starting point for potential applications to athletes.
Author Response
Dear Reviewer #5
We deeply thank you for your valuable suggestions and comments on our manuscript. Those comments are of enormous assistance to us for improving and revising our manuscript. We have studied the comments carefully and made corrections in line with the suggestions you made.
In attachment you can see our answers

Round 2
Reviewer 2 Report
Comments and Suggestions for Authors
The authors have responded correctly to all comments made.
Congrats.
Reviewer 4 Report
Comments and Suggestions for Authors
Paper has been significantly improved